# Switching from Intravenous to Subcutaneous Biological Therapy for Inflammatory Bowel Disease Patients Remains a Challenge

**DOI:** 10.3390/jcm13051357

**Published:** 2024-02-27

**Authors:** Vered Richter, Daniel L. Cohen, Ofra Kriger-Sharabi, Dana Zelnik Yovel, Nadav Kochen, Efrat Broide, Haim Shirin

**Affiliations:** 1The Gonczarowski Family Institute of Gastroenterology and Liver Disease, Shamir (Assaf Harofeh) Medical Center, Zerifin 70300, Israel; docdannycohen@yahoo.com (D.L.C.); danazelnik@gmail.com (D.Z.Y.); kochenadav@gmail.com (N.K.); haim_shirin@yahoo.com (H.S.); 2Faculty of Medicine, Tel Aviv University, Tel Aviv 6927846, Israel; efibroide@yahoo.com; 3Institute of Gastroenterology, Assuta Medical Center, Ashdod 7747629, Israel; ofranirsh@gmail.com; 4The Jecheskiel Sigi Gonczarowski Pediatric Gastroenterology Unit, Shamir (Assaf Harofeh) Medical Center, Zerifin 70300, Israel

**Keywords:** inflammatory bowel disease, biologic therapy, treatment preferences, intravenous infusion, subcutaneous, treatment cost

## Abstract

Biological inflammatory bowel disease (IBD) medications, once limited to intravenous (IV) administration, can now be administered both via IV and subcutaneously (SC). This study investigates patient preferences, willingness to switch from IV to SC, and associated factors. A questionnaire covering demographics, disease-related inquiries, quality of life, and IBD medication preferences was distributed via email, the Israeli Crohn’s Disease and Ulcerative Colitis Foundation, infusion centers, and clinics. From 454 IBD patients (median age: 42 years; 55.7% female), responses revealed a preference for SC every 8 weeks, which is comparable to daily oral dosing. Both options were significantly favored over IV every 8 weeks and SC every 2 weeks, with no statistically significant differences between the latter two. However, among patients who were experienced with both SC and IV administration, a clear preference for SC administration every 2 weeks over IV every 8 weeks surfaced. Among IV-treated patients, 54.5% resisted switching to SC. Key reasons for this included medical staff presence (57.7%), a fear of needles (46.4%), belief in infusion efficacy (37.1%), and longer intervals between infusions (36.1%). Findings suggest that transitioning from IV to SC treatment is challenging due to patient resistance, which is influenced by specific factors. Identifying and addressing these obstacles is crucial for optimizing IBD management.

## 1. Introduction

In recent years, numerous medications with varying mechanisms of action have emerged for the treatment of inflammatory bowel disease (IBD), and the methods of administration have evolved as well. Some drugs, such as Janus kinase (JAK) inhibitors and sphingosine 1-phosphate (S1P) receptor modulators, are taken orally (PO) daily. Others, like Anti-interleukin agents (e.g., ustekinumab, risankizumab), are suitable only for subcutaneous (SC) administration, typically every 8 weeks. Recently, medications that were previously limited to intravenous (IV) administration exclusively, such as the infliximab biosimilar CT-P13 and vedolizumab, have become available for both IV every 8 weeks or SC every 2 weeks. Infliximab biosimilar CT-P13 and vedolizumab were introduced in Israel in March and April 2023, expanding the treatment options for patients [1,2]. As the SC administration form of these drugs was only recently introduced, most guidelines do not yet reference it. However, according to the 2024 guideline from the British Society of Gastroenterology (BSG), both administration forms are considered therapeutic options for ulcerative colitis (UC) and Crohn’s disease [3].

In light of recent publications underscoring the safety and efficacy of SC administration of medications that are suitable for IBD maintenance via both the IV and SC routes [4,5,6,7,8,9], and considering the significantly higher costs associated with IV delivery [10], it is unequivocally in the best interest of the health system to accelerate the transition toward SC administration. The current trend of starting maintenance therapy with SC regimens implies that SC delivery will soon be the primary mode of treatment. However, transitioning established patients from IV to SC poses a challenge, as a significant portion of patients prefer to maintain IV treatment. A nationwide study among IBD patients revealed that the most preferred regimens were oral intake once or twice daily and SC every 12 or 8 weeks. Nevertheless, the IV route was the first choice in 12.8% of patients [11]. In Japan, according to a survey among IBD patients, oral administration was the preferred route for most patients; however, more patients preferred in-hospital infusion or injection therapy over self-injection therapy [12]. Brunet-Houdard et al. discovered that Crohn’s patients with impaired HRQoL preferred the IV route. They attributed this preference to their likely desire for proximity to a specialized healthcare professional, enabling them to comfortably address all their questions. Nevertheless, these patients dealing with diminished HRQoL showed a reluctance to allocate additional time for treatment, possibly due to fatigue and weariness [13]. In an Italian questionnaire study involving 311 IBD patients undergoing IV biological treatment, 49.8% of respondents indicated a preference for transitioning to SC therapy. Factors such as distance from the IBD center and personal commitments influenced the approval rate for home therapy. Notably, only a quarter of patients were aware of the subcutaneous administration option for therapeutic agents [14]. In a study conducted at Nancy Hospital in France, 16.7% of vedolizumab patients and 24.5% of those treated with infliximab declined to transition from IV to SC [15].

Patient preferences are influenced by various factors, including cost [10] and the distance from the outpatient infusion facilities [13]. The Israeli health system, founded on a universal healthcare model, contains a “Health Basket”, which comprises a list of medications that are covered by national health insurance, ensuring access to essential pharmaceuticals. The patient contributes a share of the cost up to the annual deductible of approximately USD 1100. Another unique aspect of Israel is that its small size ensures close physical proximity to hospitals and outpatient facilities. Moreover, medical services are readily available.

In our ambulatory infusion facility, patients often express resistance to transitioning from IV to SC therapy, even though the latter might offer clear advantages. Given Israel’s high incidence of IBD, it is imperative to examine the factors influencing patients’ preferences within this unique healthcare landscape. Additionally, it is particularly important to conduct an economic assessment comparing the costs of IV and SC treatments in Israel, taking into account official drug prices at pharmacies, the cost of administering treatment by a medical team, and the expenses associated with lost workdays, transportation, and parking.

Therefore, we aimed to determine our patients’ preferences regarding the routes and frequencies of administering the main IBD medications in practice, in order to assess the willingness of patients currently receiving IV treatment to transition to SC administration and to investigate the factors influencing these preferences.

## 2. Materials and Methods

### 2.1. Study Population and Design

The questionnaire was distributed via email to patients diagnosed with IBD who were registered at the Gastroenterology Institute at Shamir Medical Center and to members of the Israel Crohn’s Disease and Ulcerative Colitis Foundation during the period between June 2023 and October 2023. Additionally, IBD patients who presented for a medication infusion or clinic visit and had not previously completed the emailed questionnaire online were provided with a hard copy of the questionnaire to fill out.

### 2.2. Questionnaire

The questionnaire was developed by a team of physicians from the IBD unit of the Gastroenterology Department at Shamir Medical Center and was formulated based on daily interactions with patients in routine clinical practice. The questions were crafted in the Hebrew language. The questionnaire was divided into two sections: The first, titled “Demographic and Disease Characteristics Questionnaire”, comprised 15 items. It encompassed demographic details, information about the type and severity of IBD, concurrent medical conditions, and inquiries about living arrangements and independent hospital arrivals. A specific health-related quality of life (HRQoL) questionnaire for IBD patients, the short IBD questionnaire (SIBDQ), was included and contained ten questions aimed at evaluating the patient’s quality of life [16]. The second section was the “Treatment Preferences and Experience Questionnaire”, which consisted of 10 items. It included inquiries about the patient’s current and prior medication experiences, with a focus on the mode of administration. All study participants were asked to express their preferences on a Likert scale of 1–4 (1 being least preferred and 4 being most preferred) for various drug administration regimens. The considered regimens included oral pills once a day every day, SC injections every 2 weeks, SC injections every 8 weeks, and IV infusions every 8 weeks. The questionnaire also featured questions exploring the patient’s preferences regarding a potential switch in drug administration from IV to SC, along with the rationale behind these preferences. The complete questionnaire is provided in Appendix A.

### 2.3. Ethical Considerations

Data were analyzed in accordance with the principles outlined in the Declaration of Helsinki. The study underwent review and approval by the Shamir Medical Center Institutional Review Board (Approval No. 0104-23-ASF). Informed consent was not deemed necessary by the Ethics Committee, as the questionnaire was completed anonymously, and the act of responding to the questionnaire was considered implicit consent.

### 2.4. Statistical Analyses

Categorical variables were reported as frequency and percentage. The distribution of the continuous variables was evaluated using histograms and Kolmogorov–Smirnov test. Normally distributed continuous variables were reported as median with interquartile range. Categorical variables were compared using chi test or Fischer’s exact test and continuous and ordinal variables using independent Student’s *t*-test, or Mann–Whitny test. *t*-test was used to determine a significant difference between groups at a significance level of *p* < 0.05. All statistical tests were two-sided.

To examine factors related to each route of administration, we conducted comparisons between independent groups using χ2 or Fisher’s exact tests for categorical variables, and the Kruskal–Wallis test for quantitative parameters. Statistical analysis was performed using SPSS Statistics (IBM SPSS Statistics for Windows, Version 28.0.; IBM Corp; Armonk, NY, USA).

## 3. Results

### 3.1. Baseline Characteristics of the Patients

This research study encompassed 454 IBD patients, with 55.7% being women and 17.6% reporting smoking. The median age was 42 (IQR: 29–57). The patients’ employment status varied, with 30.4% being unemployed, 53.3% in full-time jobs, and 16.3% in part-time positions. The majority (91.2%) arrived independently at their outpatient clinic and infusion center appointments. Additionally, 87.4% were living with other people. In terms of their IBD disease, 67.6% of participants had Crohn’s disease (CD), while the remaining individuals had ulcerative colitis (UC). Additionally, 49.1% reported having active disease, and 9% were currently using corticosteroids. Furthermore, 39.2% were currently receiving IV medication, 65.2% had experience with IV medication, and 51.1% had experience with SC medication. Detailed data on the baseline characteristics are provided in Table 1.

The patients were asked if they had been poorly compliant with medications in the past. If they answered yes, they were then asked in what form the medicine was taken. One hundred and ninety-one patients (37.4%) indicated prior poor medication compliance. Among those, 62.8% reported poor compliance with oral medications, 22% with SC medicines, and 15.2% with IV infusions.

### 3.2. The Preferred Routes of Medication Delivery among the Entire Study Population

All study participants were asked to express their preferences on a scale of 1–4. Figure 1 displays the results of the patients’ preferences among the treatment options.

The highest score was observed for SC every 8 weeks at 2.88, although it was nonsignificant compared to per os (PO) every day, which scored 2.82. Both routes of administration were significantly more preferred than IV every 8 weeks and SC every 2 weeks. No statistically significant difference was found between the preferences for IV every 8 weeks and SC every 2 weeks.

### 3.3. Comparing Preferences between IV Treatment Every 8 Weeks and SC Every 2 Weeks

Our findings indicate that 110 participants (24.2%) expressed no particular preference for either administration method, 187 individuals (41.2%) favored SC every 2 weeks, and 157 participants (34.6%) preferred IV every 8 weeks. Importantly, no statistically significant difference was observed between these preferences (*p* = 0.385).

Patients who preferred IV administration every 8 weeks, as opposed to those who favored SC administration every 2 weeks, were younger, with a median age of 36 years (IQR, 26–52) compared to 46 years in the other group (IQR, 31–59), (*p* = 0.001). Additionally, patients who preferred the IV route had a higher prevalence of UC and less CD (38.8% vs. 25.1%, 61.1% vs. 74.9%, respectively, *p* = 0.017). No significant difference was observed between the two groups in terms of disease activity, quality of life, employment status, independent arrival to infusion center, living with other people, and infusion process time (Appendix A).

### 3.4. Preferred Routes of Medication Delivery According to the Patients’ Prior Experience

In our study population, there were 81 patients with prior exposure to SC administration but no history of IV infusion of biological therapy. The majority of these patients (71.6%) expressed a preference for SC treatment, while 11.1% favored IV, and 18.5% had no specific preference. Conversely, among the 145 patients with prior exposure to IV but without any history of SC therapy, a significant proportion (62.1%) preferred IV treatment, with 20.0% favoring SC, and 17.9% having no preference.

For the subgroup of 151 patients who had previous experience with both SC and IV infusion, 48.3% favored SC treatment, 29.1% preferred IV, and 22% had no specific preference. In contrast, among the 77 patients with no prior experience with either SC or IV infusion, 45.5% had no preference, 35.1% preferred SC treatment, and 19.5% favored IV treatment (Figure 2).

### 3.5. Interest of Intravenous Patients in Switching to Subcutaneous Treatment

Out of the 178 individuals undergoing IV treatment, 81 (45.5%) expressed a willingness to consider a switch to SC treatment. Most patients expressed this preference due to the convenience of self-administration at home, followed by time savings and the desire not to miss work. Concerns about the risk of infection exposure were cited by 32.1%, and 25.9% mentioned a difficulty locating veins for infusion.

Of the 97 (54.5%) who preferred to continue with IV infusion, the most common reason was the presence of a nurse or medical staff during treatment (57.7%). The next most common reasons included a fear of needles, the belief that infusion is more effective, the extended intervals between infusions, and a desire to see a doctor after each infusion session. Figure 3 delineates the specific motivations behind patients’ preferences for either staying with IV treatment (Figure 3A) or opting for a switch to SC treatment (Figure 3B).

Furthermore, patients were asked to indicate the extent to which the cost of the route of administration would influence their decision to switch from IV to SC or to remain on IV. Out of the 178 patients, 89 (50%) reported “ low influence”, 42 (23.6%) answered “moderate influence”, and 47 (26.4%) reported “high influence”.

### 3.6. Comparative Cost Analysis between IV and SC Therapy

In Israel, the cost of the drugs vedolizumab and infliximab in their IV or SC forms does not change for the patient, as they are part of the “Health Basket”, and an annual participation fee of around USD 1100 applies.

We have no access to the real cost of medications in the business contracts between pharmaceutical companies and healthcare organizations. However, the calculation of pharmacy prices for private purchases of the IV infliximab biosimilar Remsima revealed an annual total cost of USD 11,808. In contrast, the pharmacy price for the SC formulation of the infliximab biosimilar Remsima is only USD 6528 per year. As for vedolizumab, the annual pharmacy price for IV treatment is USD 11,754, while the vedolizumab SC formulation totals USD 22,968 annually [17].

In the administration of an IV infusion, a payment of USD 110 is added for the infusion treatment. Additionally, the patient incurs a USD 5.50 charge for parking or for public transportation and loses a day of work. As of January 2023, the average monthly salary in Israel is USD 3252, valuing a day’s work at approximately USD 162. A detailed breakdown of the costs is provided in Appendix A.

## 4. Discussion

Recently, medications that were previously limited to IV administration only have become available for both IV every 8 weeks or SC every 2 weeks. However, in our study, we found that 54.5% of individuals undergoing IV treatment expressed a preference to continue with IV infusion rather than switching to SC administration. The primary reason why patients receiving IV infusions refused to switch was the presence of a nurse or medical staff during the treatment (54.5% of the 178 patients). Other influential factors included a fear of needles, the belief that infusion is more effective, extended intervals between infusions, and a desire for consultations with a doctor after each treatment. This aligns with findings from a previous study in France, where patients resisted switching from IV to SC due to concerns about a potential loss of efficacy, a more spaced-out medical follow-up, increased administration frequency, and the introduction of self-administered injections [15].

A significant finding in our study was the belief held by 37.1% of participants that infusion is inherently more effective than SC administration. However, this assertion lacks support in the existing literature. Contrary to this perception, the multicenter, prospective SECURE study demonstrated that switching from IV to SC infliximab in patients with IBD in remission is safe and resulted in higher infliximab concentrations at 16 weeks in both UC and CD patients [18]. SC infliximab has been shown to maintain consistently higher levels of the drug and exhibit significantly lower immunogenicity [10,19], excluding those receiving IV infliximab at 10 mg/kg every four weeks [9]. Significant as well as level elevation was also documented with SC vedolizumab following such a switch [10]. Cerna et al. proposed that SC infliximab may be a “Biobetter”, aiming to improve upon existing biologics therapies. They suggested that, due to its low immunogenicity, SC infliximab can be used to reinduce therapy in refractory CD patients with positive neutralizing anti-infliximab antibodies [8]. Other real-world data further substantiate the evidence for both infliximab and vedolizumab [6,7,20].

In Finland, economic factors were the main reason for IBD patients not switching from IV to SC therapy [10]. Contrary to Finland, where SC infliximab and vedolizumab are not fully reimbursed, in Israel, the patient’s deductible for the drug is the same price for SC and IV. However, when our patients were asked whether the extent to which the cost of the route of administration would influence their decision to switch from IV to SC, only 26.4% reported a “high influence”.

The total infusion process time showed no influence on patients’ preferences, encompassing the time to the infusion center and back, as well as the time needed for IV administration. Previous research indicates that shortening the infusion time is possible. Moreover, shorter, one-hour nurse-led infusion protocols revealed a lower incidence of infusion reactions compared to the standard two-hour procedure [21]. Napolitano et al. found a noteworthy correlation between the approval rate for SC therapy and the distance from the IBD center [14]. Intriguingly, within the initially investigated range of 50 km, the distance does not seem to exert a substantial impact on the decision. This may explain our results, as Israel is characterized by its small size and close physical proximity to hospitals and outpatient facilities.

Regarding the preference to switch from IV to SC, reasons mentioned included convenient self-administration at home, the duration spent at the infusion unit, and the consequential impact on work commitments. Additionally, about a third of those who preferred to switch from IV to SC feared exposure to infections in the hospital, and about a quarter faced difficulty accessing veins. In addition to the general preference for SC over IV drug administration among both patients and healthcare providers, the Virtual Community Meeting, which features insights and opinions from active IBD-treating healthcare professionals, emphasizes the risk associated with treatment non-compliance [22].

In our study, the most favored patient preferences for treatment methods, assuming a hypothetical scenario where drugs could be administered in various manners, was SC administration every 8 weeks, although it did not significantly differ from daily PO administration. Both SC every 8 weeks and PO every day were significantly more favored than IV treatment every 8 weeks and SC every 2 weeks. In a recent nationwide study conducted in France with 1850 participants, it was found that the oral route was the preferred method of medication delivery. The research also highlighted that the acceptability of treatment is significantly influenced by the administration rhythm. Notably, in patients with inflammatory bowel disease (IBD), subcutaneous (SC) treatment with longer intervals between injections (≥8 weeks) emerged as one of the most widely accepted modalities. [11]. It is noteworthy that our study corroborates these findings, even when patients have no financial considerations, and when infusion facilities are readily available and in close proximity. In a review of 31 studies comparing IV infusion and SC injection for non-immunoglobulin therapies, preferences varied: twenty studies favored SC administration, seven favored IV infusion, and four showed no overall preference [23].

We observed no statistically significant differences in preferences between IV administration every 8 weeks and SC every 2 weeks, as vedolizumab and infliximab currently offer dual administration methods with these respective regimens.

While delineating patient profiles for each administration route, our study stands out by considering disease-related factors alongside additional variables, including quality of life, employment status, independence during infusion center visits, and travel time from home. We observed that those favoring SC every 2 weeks were younger and had a higher prevalence of CD. However, in other studies, age was not a predictor for willingness to switch [15,24]. Contrary to intuition and a previous study by Brunet-Houdard et al. [13], our results did not support the notion that patients with more severe illnesses or reduced quality of life might prefer IV treatment, potentially due to the additional interaction with healthcare professionals during infusions. Additionally, our hypothesis that individuals working full-time or who are unable to visit the infusion center independently would prefer SC treatment was not confirmed by our findings.

Our data correlated well with previous data indicating that prior therapeutic experiences influence patient preferences to the known and familiar [11,23]. K. van Deen et al. conducted a study involving 1077 participants with IBD. The survey revealed that 49% preferred an SC medication administered every 2 weeks, while 51% favored an intravenous IV medication every 8 weeks. They found that current and past medication use strongly shaped SC/IV preferences. Specifically, among patients with experience only with SC administration, 61% preferred SC, while among those with experience only with IV administration, 40% favored SC. For patients with experience with both, 53% expressed a preference for SC administration [25]. Indeed, in our study, patients with prior SC exposure but no IV history preferred SC treatment every 2 weeks, while those with prior IV exposure but no SC history favored the IV route every 8 weeks. However, notably, among those with experience with both methods, a clear preference for SC administration every 2 weeks surfaced. These findings indicate that the SC route of administration is likely to be more favorable for most patients upon trial.

Due to confidential agreements between each HMO and pharmaceutical companies, an accurate comparison of IV and SC prices is not feasible. However, private pharmacy prices indicate that annual costs for infliximab biosimilar Remsima infusion are nearly double those for SC treatment. Infusion procedures and their associated indirect costs, such as lost workdays and transportation costs, further increase the overall infusion treatment expenses. Opting for Remsima SC administration in Israel facilitates a substantial cost reduction and allocates resources for other necessary medical interventions. In contrast, for vedolizumab, our calculations revealed that the SC administration costs are almost double those of the IV formulation and administration.

Our study has some limitations. First, this is a questionnaire-based study which introduces the risk of social desirability bias, volunteer bias among patients enrolled, and limited depth of information due to predefined response options. However, in our specific case, the use of questionnaires was deemed necessary, since we sought information about patients’ preferences, which is not recorded in patient files. In addition, we distributed questionnaires to patients in infusion centers and medical clinics to ensure the inclusion of individuals who may face challenges in using the internet. However, this likely resulted in an over-representation of patients who are treated with IV administration compared to typical practice. Additionally, the study was conducted only in Israel, and certain aspects may not be applicable to other countries. Finally, we did not use objective markers to assess IBD activity.

## 5. Conclusions

No significant preference differences were noted between IV every 8 weeks and SC every 2 weeks, which is consistent with the available dual-administration drugs, and 54.5% resisted the switch from IV to SC. However, among those acquainted with both methods, a preference emerged for SC administration every 2 weeks, hinting at its potential superiority upon trial. While not medically justified, switching patients from IV to SC treatment remains a significant challenge. Understanding and clarifying the major obstacles of patients in each country may assist in overcoming this resistance. These results underscore the crucial role of guiding healthcare personnel in skillfully communicating and facilitating patients’ transitions from IV to SC. Further qualitative studies, such as in-depth interviews or focus groups, can be considered to investigate psychosocial factors contributing to resistance, thereby enhancing our understanding and improving the transition from IV to SC administration.

## Figures and Tables

**Figure 1 jcm-13-01357-f001:**
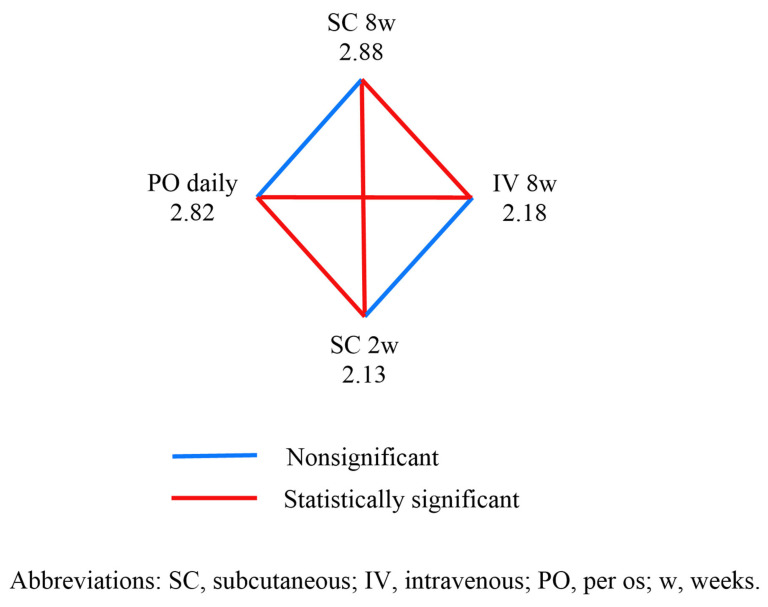
Comparison of patients’ preferences among the main treatment regimens.

**Figure 2 jcm-13-01357-f002:**
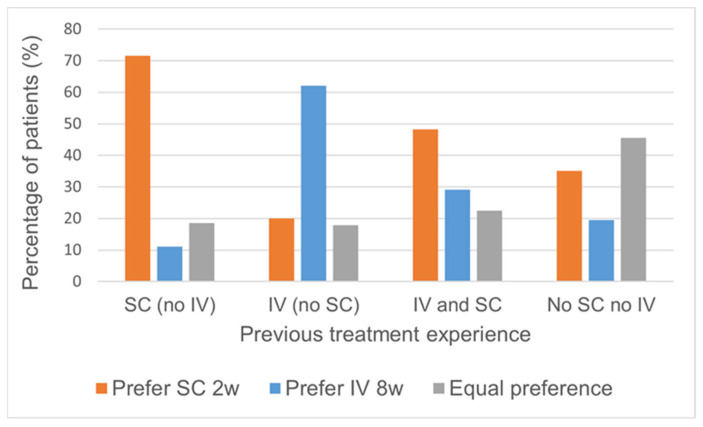
Patients’ preferred medication delivery routes based on prior experience.

**Figure 3 jcm-13-01357-f003:**
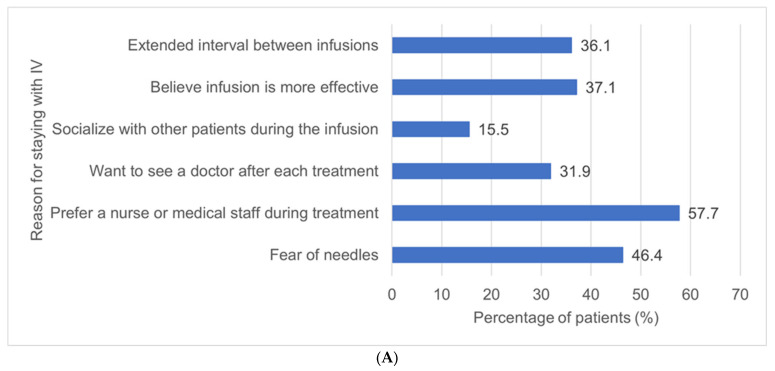
Reported reasons for preferring remaining on IV administration (**A**) and switching from IV to SC (**B**).

**Table 1 jcm-13-01357-t001:** Baseline characteristics of study population.

Characteristics	IBD Patients *n* = 454
Age (median, (IQR))	42 (29–57)
Female gender (*n*, %)	253 (55.7)
Smoking (*n*, %)	80 (17.6%)
BMI (median, (IQR))	24.4 (21.4–28.0)
Employment status	
Currently not employed (*n*, %)	138 (30.4)
Full-time job (*n*, %)	242 (53.3)
Part-time job (*n*, %)	74 (16.3)
Independent arrival to infusion center (*n*, %)	414 (91.2)
Living with other people	397 (87.4)
IBD type	
Crohn’s disease (*n*, %)	307 (67.6)
Ulcerative colitis (*n*, %)	147 (32.4)
Age at diagnosis (median, (IQR))	25 (18–39)
Active disease (*n*, %)	223 (49.1)
Current use of corticosteroids (*n*, %)	41 (9)
Another systemic disease ^1^ (*n*, %)	165 (36.3)
Current IBD treatment ^2^	
Per os (*n*, %)	132 (29.1)
Subcutaneous (*n*, %)	232 (51.1)
Intravenous (*n*, %)	296 (65.2)
Poor compliance with medication intake (*n*, %)	191 (42.1)
Per os (*n*, %)	120 (62.8)
Subcutaneous (*n*, %)	42 (22.0)
Intravenous (*n*, %)	29 (15.2)

Abbreviations: IBD, inflammatory bowel disease; IQR, interquartile range. ^1^ Another systemic disease included lung, liver, kidney, and ischemic heart disease; congestive heart disease; peripheral vascular disease; diabetes mellites; cerebrovascular disease; rheumatic diseases; and peptic ulcer disease. ^2^ The values exceed 100% due to multiple treatments for some patients.

## Data Availability

The raw data supporting the conclusions of this article will be made available by the authors on request.

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
