# Peer review of "Switching from Intravenous to Subcutaneous Biological Therapy for Inflammatory Bowel Disease Patients Remains a Challenge"

_jcm, 2024, doi:10.3390/jcm13051357_

Round 1

Reviewer 1 Report

Comments and Suggestions for Authors

This paper describes the results of a study aimed at understanding patient preferences, willingness to switch from IV to SC, and eventual associated factors. Data were collected by a questionnaire covering demographics, disease-related inquiries, quality of life, and IBD medication preferences distributed via email to patients diagnosed with IBD members of the Israeli Crohn’s Disease and Ulcerative Colitis Foundation or registered at infusion centres or clinics.

Unfortunately, the methodology regarding the development, validation or testing of the questionnaire (i.e. interviews with stakeholders, focus groups, Delphi, etc.) is missing. If none of these steps was performed before its distribution, this could probably be the cause of some weaknesses of this work, i.e. the time needed for EV administration or the time spent in the hospital is not considered in this study (but could be important, please look i.e. at https://doi.org/10.1016/j.dld.2015.01.152) as well as safety issues or concerns or anything concerning the reassessment of the disease and IBD patient care. Medication adherence, which could be a trigger point regarding willingness to switch, is assessed without a standardised and validated scale (i.e. the MMAS-8).

Regarding the results, the response rate would be very important but is missing.

Regarding the discussion some results mentioned in this chapter are not clearly described in the previous chapter or the supplementary material.

Author Response

Thank you for reviewing our article. We appreciate your valuable feedback, and we have addressed each of your points in detail.

The response is attached

Reviewer 2 Report

Comments and Suggestions for Authors

Dear Authors

Although your paper is well written it is not innovative and does not add any useful information for physicians. This topic has already been studied and data presented only concern Slovenian population, while in literature data from multicentre studies are available. Probably data about the comparison of efficacy between IV and SC formulation of biologics may be more interesting and useful.

Author Response

(The authors gave the same response as above.)

Reviewer 3 Report

Comments and Suggestions for Authors

This is an excellent audit based on a questionnaire to elicit preferences of drug delivery for patients with IBD.

Patients established on oral medication were not included.

Responses were received from 454 patients but how many were invited. 

The most preferred route was subcutaneous every 8 weeks. This was not significant from the oral route. Were they given a choice of oral medications. 

Cost is of interest from a tax payer point of view but admittedly the investigators did not have full access to this information. I note that the cost was not that important to the patient as they obtained the drug free of charge.  

Author Response

(The authors gave the same response as above.)

Reviewer 4 Report

Comments and Suggestions for Authors

The manuscript entitled “Switching from Intravenous to Subcutaneous Biological Therapy for Inflammatory Bowel Disease Patients: Still a Challenge” discusses a relevant topic is of huge interest regarding the treatment preference of the disease. It is novel within the literature, interesting and is generally well described and conducted.

Although this is a well-studied and well-written manuscript few suggestions for improving the manuscript are as follows:

1.    Please address how the treatment guideline of IBD defines the choose/preference for mode of treatment.

2.    The clear preference is SC every 8 weeks, but the article fails to explain whether the condition is only hypothetical or treatment options with SC every 8 weeks is really available. All results/discussion (table S3, Fig. 2 and Fig.3) are based on comparison between IV every 8 weeks and SC every 2 weeks. Please elaborate table S3 to include the treatment regimens, if available.

3.    Please provide full form in line 47 (ACCEPT2).

Author Response

(The authors gave the same response as above.)

Round 2

Reviewer 1 Report

Comments and Suggestions for Authors

Thanks for providing a revision of the original manuscript, this version is significantly improved.